# Effects of Fermented Milk Containing *Lacticaseibacillus paracasei* Strain Shirota on Constipation in Patients with Depression: A Randomized, Double-Blind, Placebo-Controlled Trial

**DOI:** 10.3390/nu13072238

**Published:** 2021-06-29

**Authors:** Xiaomei Zhang, Shanbin Chen, Ming Zhang, Fazheng Ren, Yimei Ren, Yixuan Li, Ning Liu, Yan Zhang, Qi Zhang, Ran Wang

**Affiliations:** 1Department of Nutrition and Health, China Agricultural University, Beijing 100091, China; bh2020270136@cau.edu.cn (X.Z.); chenshanbin@cau.edu.cn (S.C.); renfazheng@cau.edu.cn (F.R.); liyixuan@cau.edu.cn (Y.L.); dadaliu@cau.edu.cn (N.L.); zq809425044@swu.edu.cn (Q.Z.); 2School of Food and Chemical Engineering, Beijing Technology and Business University, Beijing 100048, China; zhangming@th.btbu.edu.cn; 3Key Laboratory of Functional Dairy, Co-Constructed by Ministry of Education and Beijing Government China Agricultural University, Beijing 100083, China; renyimei123@cau.edu.cn; 4Hebei Engineering Research Center of Animal Product, Sanhe 065200, China; zhangyan@cau.edu.cn

**Keywords:** *Lacticaseibacillus paracasei* strain Shirota (LcS), constipation, depression

## Abstract

Probiotics have been shown to benefit patients with constipation and depression, but whether they specifically alleviate constipation in patients with depression remains unclear. The aim of this study was to investigate the effect of *Lacticaseibacillus paracasei* strain Shirota (LcS), formerly *Lactobacillus casei* strain Shirota, on constipation in patients with depression with specific etiology and gut microbiota and on depressive regimens. Eighty-two patients with constipation were recruited. The subjects consumed 100 mL of a LcS beverage (10^8^ CFU/mL) or placebo every day for 9 weeks. After ingesting beverages for this period, we observed no significant differences in the total patient constipation-symptom (PAC-SYM) scores in the LcS group when compared with the placebo group. However, symptoms/scores in item 7 (rectal tearing or bleeding after a bowel movement) and items 8–12 (stool symptom subscale) were more alleviated in the LcS group than in the placebo group. The Beck Depression Index (BDI) and Hamilton Depression Rating Scale (HAMD) scores were all significantly decreased, and the degree of depression was significantly improved in both the placebo and LcS groups (*p* < 0.05), but there was no significant difference between the groups. The LcS intervention increased the beneficial *Adlercreutzia*, *Megasphaera* and *Veillonella* levels and decreased the bacterial levels related to mental illness, such as *Rikenellaceae_RC9_gut_group*, *Sutterella* and *Oscillibacter.* Additionally, the interleukin (IL)-1β, IL-6, and tumor necrosis factor-α (TNF-α) levels were significantly decreased in both the placebo and LcS groups (*p* < 0.05). In particular, the IL-6 levels were significantly lower in the LcS group than the placebo group after the ingestion period (*p* < 0.05). In conclusion, the daily consumption of LcS for 9 weeks appeared to relieve constipation and improve the potentially depressive symptoms in patients with depression and significantly decrease the IL-6 levels. In addition, the LcS supplementation also appeared to regulate the intestinal microbiota related to mental illness.

## 1. Introduction

Depressive disorder, also known as depression, is a type of mood disorder characterized by persistent low moods. The main symptoms are long-term depression, lack of motivation, loss of pleasure, and physical dysfunction. According to the World Health Organization (WHO), the global incidence of depression is approximately 4.4% [1]. Notably, stress, dietary habits, and physical inactivity have been implicated in depression etiology [2].

Correlations have been shown to exist between depression and gastrointestinal disease. Moreover, individuals with intestinal discomfort have a higher risk of depression. In a survey involving 18,571 subjects, those with gastrointestinal discomfort had a higher prevalence of depression than the general population (7.5% versus 2.9%), and the depression prevalence in the population with at least two gastrointestinal discomfort issues was as high as 13.4% [3]. A meta-analysis of 7179 constipation-type irritable bowel syndrome (IBS-C) patients and 69,989 chronic idiopathic constipation (CIC) patients from 35 studies reported that the incidence of depression in both groups was up to 12.5–69.2% and 14.6–29.2%, respectively [4]. Almost all cases of depression are accompanied by gastrointestinal symptoms, and 83.5% of patients with depression experience diarrhea and constipation symptoms, which are the most common gastrointestinal symptoms [5]. These data suggest the gut–brain axis, or the gut and brain interaction pathway, may be involved in the development and progression of depression. Similarly, gastrointestinal symptoms, including constipation, can impair the quality of life of patients with depression.

Depression pathogenesis involves dysfunctional elements in neuroendocrine, immune responses, synaptic and plasticity functions, and neurogenesis pathways. Growing evidence suggests that the gut microbiota play key roles in the development of depression, and several studies on this subject are currently underway. Research has indicated that the gut microbiota in patients with depression differs from the gut microbiota of healthy individuals. Aizawa et al. reported that the *Bifidobacterium* levels in patients with depression were significantly lower than in healthy people and that the total number of bacteria in lactobacilli was also lower [6]. Kunugi et al. reported results suggesting that individuals with lower bacterial counts for *Bifidobacterium* and/or *Lactobacillus* are more common in patients with major depressive disorder than in healthy controls [7]. Studies on the gut–brain axis have also been conducted to determine the effects of gut microbiota on depression. Germ-free (GF) animal studies have reported that the gut microbiota exert functions in basic neurogenic processes, including the formation of the blood–brain barrier, myelination, neurogenesis, and microglia maturation, which, in turn, affect animal behaviors [8]. In another work, mice that experienced maternal deprivation exhibited anxiety and depressive behaviors, abnormalities in corticosterone stress hormone levels, and intestinal dysfunction [9]. Additionally, GF mice experienced abnormal stress hormone levels and intestinal dysfunction after maternal deprivation but no anxiety and depression-like behaviors [9].

Probiotics effectively regulate gut microbiota and alleviate gastrointestinal diseases such as diarrhea and constipation. A recent meta-analysis of 1349 subjects involved in 10 probiotic interventions towards depression revealed that probiotics effectively alleviated mild and moderate depressive symptoms [10]. The *Lacticaseibacillus paracasei* strain Shirota (LcS), formerly *Lactobacillus casei* strain Shirota, has a history of over 80 years of safe consumption and proven health benefits and is supported by extensive scientific research on the reduction of functional and infectious gut diseases and immune-modulating effects.

The effectiveness of LcS in alleviating constipation has also been demonstrated in several studies. The consumption of LcS beverages improved the constipation severity and stool consistency and accelerated the colonic transit in patients with chronic constipation [11,12]. LcS also improved the stool consistency in healthy individuals, patients with Parkinson’s disease, women during puerperium, and gastrectomized patients [13,14,15,16,17]. Hence, LcS may have the potential to ameliorate constipation in patients with depression.

LcS also improved hard stools during constipation, possibly by increasing butyrate-producing bacteria and the subsequent stool butyrate levels by 39.7% [18]. Several studies have demonstrated that LcS improved the bowel movement frequency and stool quality, thereby alleviating constipation [11,12,13,14,15,16,17]. A recent study revealed that LcS relieved stress-induced anxiety and decreased salivary cortisol levels, abdominal dysfunction, sleep disturbance, and other physical symptoms and preserved the gut microbiota of stressed healthy populations [19,20,21,22]. However, whether LcS exerts an impact on depressive symptoms remains unclear.

The primary aim of this study was to investigate the effects of LcS on constipation in patients with depression. LcS has been demonstrated as an effective treatment for constipation in various populations. Therefore, we sought to investigate such effects in patients with depression who may also have a specific etiology and gut microbiota. As a secondary objective, we investigated the effects of LcS on depressive symptoms.

## 2. Material and Methods

### 2.1. Study Design

This study was a two-arm parallel design, randomized, double-blind placebo-controlled trial (RCT). The required sample size was estimated based on Beck Depression Index (BDI) scores from Akkasheh et al. [23], considering an α-error of 0.05 and β-error of 0.20. Thirty subjects were required per group, yet 82 subjects were recruited to compensate for the probable loss to follow-up.

### 2.2. Ethical Considerations and Inclusion/Exclusion Criteria

The study protocol was approved by the Ethics Committee of China Agricultural University (research project identification No. CAUHR-2019014 on 3 September 2019). This trial was registered at http://www.chictr.org.en (accessed on 16 September 2019) as ChiCTR1900025972. Subjects who signed the informed consent form were screened for eligibility according to the inclusion and exclusion criteria.

Inclusion criteria: individuals aged 18–60 years at the beginning of the study; body mass index (BMI) 18.5–29.9; depression as diagnosed by psychiatrists according to the diagnostic criteria for depressive episodes in the American Diagnostic and Statistical Manual of Mental Disorders (5th Edition) (DSM-5); and a constipation diagnosis according to the Rome IV Criteria [24], including two or more symptoms of the following: (i) straining more than 25% of defecations, (ii) lumpy or hard stools in more than 25% of defecations, (iii) sensation of incomplete evacuation in more than one-fourth (25%) of defecations, (iv) sensation of anorectal obstruction/blockage in more than one-fourth (25%) of defecations, (v) manual maneuvers to facilitate more than one-fourth (25%) of defecations, and (vi) fewer than three spontaneous bowel movements per week. In addition, a Hamilton depression rating scale (HAMD) 17 items score of ≥8 is considered as depression [25].

Exclusion criteria: the use of systemic antibiotics or antimycotics in the 30 days prior to the study; use of antidiarrheal or laxative medication in the 30 days prior to the study; investigator uncertainty about the willingness or ability of subjects to comply with the protocol requirements; persons with a milk protein allergy, lactose intolerance, or constipation issues of organic or neurological origin; pregnant or breastfeeding women; the use of antidepressants within two weeks prior to the study; and subjects in any other studies within two months prior to entry into this study.

### 2.3. Data Collection and Study Timeline

At the screening visit, demographic data, eligibility criteria, medical history, and concomitant medications were collected. All data, except medical history, were obtained from questionnaires filled in by investigators during screening visits. The subjects were evenly randomized into the LcS or placebo group (1:1) stratified by medication status. Randomization was conducted by Dr. Guoshuang Feng’s team (China National Center for Children’s Health). The subjects were randomized using a computer-generated block randomization process to two groups in equal proportions using the SAS program version 9.4 (SAS Institute Inc., Cary, NC, USA).

The test drinks were a fermented dairy beverage and a placebo. Each 100-mL bottle of the fermented dairy beverage contained at least 1.0 × 10^10^ CFU of *Lacticaseibacillus paracasei* YIT 9029 (strain Shirota: LcS), water, sugar, skimmed milk powder, glucose, and flavor. The nutrient composition of one bottle was 287 kJ of energy, 1.2 g of protein, 0 g of lipid, 15.7 g of carbohydrates, and 19 mg of sodium. The placebo contained the same components as the fermented dairy beverage but did not contain any bacteria. The taste of the placebo was made similar to that of a fermented dairy beverage by adding lactic acid. The taste, flavor, and external appearance of the placebo were indistinguishable from those of the fermented dairy beverages. The fermented dairy beverage and the placebo were both manufactured and distributed by Yakult Corporation, Shanghai, China. They were refrigerated until ingestion.

The study consisted of a 14-day baseline period, a 9-week ingestion period, and four visits (V1–V4, 21 days apart) (Figure 1). During the baseline period, subjects continued their normal diet but excluded dairy fermented products. At V1, V2, and V3, subjects were given 21 bottles of fermented dairy beverage or placebo. The subjects were instructed to drink one bottle after every lunch and fill in a daily diary. This comprised questions on the study product intake (only for the 9-week ingestion period); other food intake; defecation frequency; stool consistency; constipation severity; constipation-related symptom scores; received medication; and any other symptoms of discomfort (diarrhea, vomiting, illness, etc.). At V1 and V4, fecal and blood samples were collected. The BDI and HAMD indices were used to assess the degree of depression. Degree of depression according to BDI scores: 0–4, No; 5–7, mild; 8–15, moderate; and >15, severe [26] and degree of depression according to HAMD 17 item scores: 0–7, No; 8–17, mild; 18–24, moderate; and >24, severe [25]. PAC-SYM scores were used to assess the constipation severity. The 12-item PAC-SYM was divided into three symptom subscales: abdominal (items 1–4), rectal (items 5–7), and stool (items 8–12). Items were scored on a 5-point Likert scale, with scores ranging from 0–4 (0 = “symptom absent”, 1 = “mild”, 2 = “moderate”, 3 = “‘severe”, and 4 = “very severe”) [27]. The primary outcomes were changes in the PAC-SYM scores. The secondary outcomes included changes in the BDI and HAMD scores, intestinal microbiota, and serum parameters.

### 2.4. Changes in Intestinal Microbiota Composition

Microbiota compositional analyses were performed on the V3–V4 region of the 16S rRNA gene using Illumina sequencing (Majorbio Bio-Pharm Technology Co., Ltd., Shanghai, China) [28]. Briefly, DNA was extracted from stool samples according to the manufacturer’s protocols using the E.Z.N.A. soil DNA kit (Omega Bio-Tek, Norcross, GA, USA). The V3–V4 region of the bacterial 16S rRNA gene was amplified by polymerase chain reaction (PCR) [18]. The primers 338F (5′-barcode-ACTCCTACGGGAG-GCAGCAG-3′) and 806R (5′-GGACTACHVGGGTWTCTAAT-3′) were used. The PCRs were conducted using the following program: initial 95 °C for 3 min, followed by 27 cycles at 95 °C for 30 s. 55 °C for 30 s, 72 °C for 45 s, and a final extension at 72 °C for 10 min. Amplicons were extracted from 2% agarose gels, purified (using the AxyPrep DNA Gel Extraction Kit (Axygen Biosciences, Inc, Union City, CA, USA)), pooled in equimolar concentrations, and paired-end sequenced (2 × 250) on an Illumina MiSeq platform (Illumina, Inc, San Diego, CA, USA) according to the standard protocols. Raw reads were deposited into the National Center for Biotechnology Information Sequence Read Archive database. Raw fastq files were demultiplexed and quality-filtered using Quantitative Insights Into Microbial Ecology (QIIME) software. Operational Taxonomic Units (OTUs) were clustered with a 97% similarity cut-off using UPARSE and chimeric sequences identified and removed using UCHIME analyses. Sequences aligned using ClustalW2 were used to construct a neighbor-joining tree with R package ape. The tree and OTU abundance were then used to calculate weighted UniFrac distances with the R package GUniFrac. OTUs with proportional abundances of at least 0.1% in at least three samples were retained for the downstream analysis. The taxonomy of each 16S rRNA gene sequence was analyzed by RDP Classifier against the Silva (SSU132) 16S rRNA database using a 70% confidence threshold [29].

The relative abundance of different genera in each sample was calculated and compared between visits and groups using repeated ANOVA measurements using R package stats (3.3.1). Chao and Shannon indices were calculated to assess the alpha diversity. Principal coordinate analysis (PCoA) based on Bray–Curtis distances of OTUs provided an overview of the gut microbial differences between groups. Analysis of similarity (ANOSIM) was used to compare weighted UniFrac distances between and within visits. Weighted UniFrac is a quantitative measure of diversity that detects changes in how many sequences from each lineage are present, as well as detecting changes in which taxa are present. Detection of differentially abundant taxa between the two groups was done using Linear discriminant analysis (LDA) Effect Size (LEfSe) [30], and LDA values ≥ 2 with a *p*-value < 0.05 were considered significantly enriched.

### 2.5. Serum Parameters 

Blood was collected by a qualified nurse and centrifuged at 3500 rpm for 10 min at 4 °C. Serum was collected and stored at −80 °C until assessment. The serum inflammatory factors interleukin (IL)-1β, IL-6, and tumor necrosis factor-α (TNF-α) were measured by a commercial enzyme-linked immunosorbent assay (ELISA) kit (ExCell Bio, Shanghai, China) as per the manufacturer’s recommendations.

### 2.6. Statistical Methods

In addition to fecal intestinal microbiota analyses, data were reported as the mean ± standard deviation (SD) unless otherwise indicated. Statistical analyses were performed using SPSS Version 21 (IBM Corp, Armonk, NY, USA). A chi-square test was performed for gender, and *t*-tests were used for age, weight, and BMI analyses. The Wilcoxon signed-rank test was used to compare data between visits, and the Wilcoxon rank-sum test was used to compare groups for constipation-related symptom, PAC-SYM, and depression scores and, also, serum parameters. *p* < 0.05 was considered an indication of statistically significant differences.

## 3. Results

### 3.1. Subject Information

Eighty-two (82) subjects were recruited following the inclusion criteria, 69 subjects completed the study (Figure 2), and the study compliance was 84%. No adverse events were observed. Basic subject information, including age, weight, BMI, HAMD indices, and gender, are shown (Table 1). No significant differences were observed between the groups in terms of age, weight, BMI, HAMD, or gender.

### 3.2. Constipation-Related Symptom Scores

After 9 weeks of product ingestion, a significant reduction (*p* < 0.05) was observed in the total PAC-SYM scores: from 14.39 ± 9.67 to 5.87 ± 6.05 in the LcS group and from 15.00 ± 10.39 to 8.97 ± 7.87 in the placebo group (Table 2). We observed no significant differences in the total PAC-SYM scores between groups (*p* = 0.055). After ingesting LcS for 9 weeks, the symptoms in items 1–12 of the PAC-SYM were significantly alleviated, but in the placebo group, only the symptoms in items 1, 6, 7, and 9–12 were significantly alleviated (*p* < 0.05). In addition, the symptoms in item 7 and the stool symptom subscale (items 8–12) were more significantly alleviated in the LcS group when compared to the placebo (*p* < 0.05). These results indicated that the constipation symptoms improved after ingesting LcS when compared with the placebo over 9 weeks.

### 3.3. Depression-Related Symptom Scores

At baseline, the subjects had significantly higher BDI and HAMD scores than at the end of the ingestion period. At the end of ingestion period, the BDI and HAMD scores were all significantly decreased, and the degrees of depression were significantly improved in both groups (*p* < 0.05, Table 3). These results showed that the LcS and placebo helped relieve depression.

### 3.4. Changes in Intestinal Microbiota Composition

The α-diversity of the gut microbiota at the OTU level was identified, and the Chao index or Shannon index was not significantly different among the two groups (Figure 3A,B).

The β-diversity of the gut microbiota at the OTU level was identified using PCoA based on Bray–Curtis distances. The overall microbial composition was observed between both groups (Figure 3C–E). However, the difference between the V1 and V4 of the LcS group and V4 of the LcS group and placebo group were smaller, indicating that LcS did not alter the gut microbiota β-diversity.

The most abundant phyla were *Firmicutes*, *Bacteroidetes*, *Proteobacteria* and *Actinobacteria* (Figure 4). There were no significant differences between the groups at V4.

The gut microbiota composition at the genus level was also analyzed (Figure 5). Bacteria with a relative abundance of more than 1% were shown at the genus level (Figure 5A). Wilcoxon rank-sum tests were performed to compare differences in the fecal bacterial communities between the two groups at the genus level. When compared to the baseline, the relative abundance of *Adlercreutzia* and *Megasphaera* was significantly increased after the LcS intervention for 9 weeks (Figure 5B), and the relative abundance of *Klebsiella*, *Tyzzerella_4* and *Eggerthella* was significantly decreased after the placebo intervention for 9 weeks (Figure 5C). When compared to the placebo (Figure 5D), the relative abundance of *Veillonella* was significantly increased, and *Rikenellaceae_RC9_gut_group*, *Sutterella* and *Oscillibacter* were significantly decreased after the LcS intervention for 9 weeks. LEfSe was used to further determine whether specific bacterial taxa were differentially enriched in V4 of the LcS group and placebo group. Using a logarithmic LDA score cut-off of 2, 17 discriminatory genera were identified as key discriminants (Figure 5E). *Veillonella*, *Neisseria*, *Ralstonia* and *Eggerthella* were significantly overrepresented in the feces of the LcS group, whereas *Rikenellaceae_RC9_gut_group*, *Sutterella*, *Oscillibacter* and *Rothia* were enriched in the placebo group.

### 3.5. Serum Pro-Inflammatory Cytokine Levels

The serum IL-1β, IL-6, and TNF-α concentrations were investigated to assess the inflammation (Table 4). The IL-1β, IL-6, and TNF-α levels displayed no significant differences between the placebo and LcS groups at the baseline. After 9 weeks of the LcS intervention, the IL-1β, IL-6, and TNF-α were significantly decreased. Additionally, the IL-6 levels were significantly lower in the LcS group than the placebo at the end of the ingestion period.

## 4. Discussion

In this study, we confirmed that constipation symptoms, especially stool symptoms evaluated with PAC-SYM, were alleviated by LcS. Additionally, LcS significantly decreased the BDI and HAMD scores, although changes in the depressive symptoms were not significant between groups. Furthermore, LcS significantly decreased the IL-6 levels and regulated the intestinal microbiota associated with mental illness. To our knowledge, this is the first probiotic double-blind RCT to evaluate constipation symptoms in patients with depression.

LcS is a widely used probiotic strain, and our finding that constipation was alleviated by LcS was consistent with previous studies [11,12,13,14,15,16,17]. The depression-alleviating function of LcS was also confirmed, although changes in the BDI and HAMD scores were not significantly different between the groups. Similarly, previous evidence indicated that antidepressant benefits for the treatment of depression were due to the placebo response [31]. An RCT of 423 women in the postpartum period reported with subject depression scores above the cut-off point were not significantly different between the probiotic treatment and placebo groups [32]. Another reason for their depressive states (the BDI and HAMD scores) improving significantly in the placebo group was the placebo containing milk, and it was reported that positive health and mood changes were associated with the consumption of raw milk in a consumer survey [33].

The gut microbiota are essential to health and have recently become a target for live bacterial cell biotherapies for various chronic diseases, including functional constipation and depression [34,35]. In our study, many taxa were found to be differentially abundant between the LcS and placebo groups, as identified by the Wilcoxon rank-sum and LEfSe. When compared to the baseline, the relative abundance of *Adlercreutzia* and *Megasphaera* was significantly increased after the LcS intervention for 9 weeks. *Adlercreutzia* is an equal-producing bacterium, exhibits beneficial effects, and possesses immunoregulatory properties [36,37]. *Megasphaera* produces butyric acid [38]. When compared to the placebo, the relative abundance of *Veillonella* was significantly increased, and *Rikenellaceae_RC9_gut_group*, *Sutterella* and *Oscillibacter* significantly decreased after the intervention with LcS for 9 weeks. *Veillonella* uses lactate and metabolizes it to weaker acids, such as acetate and propionate. In a recent clinical study to investigate the effects of a LcS probiotic drink on dental plaque microbiota, the relative abundance of *Veillonella* was significantly increased following a 28-day LcS intervention [39]. *Rikenellaceae* is associated with schizophrenia [40]. *Sutterella* is a widely prevalent commensal with a mild proinflammatory capacity in the human gastrointestinal tract and is associated with several conditions, such as type 1 diabetes and inflammatory bowel disease [41]. In addition, it is a major microbiota component in children with autism and gastrointestinal disturbances [42]. The *Oscillibacter*-type strain produces valeric acid as its main metabolic end product, a homolog of neurotransmitter gamma-aminobutyric acid (GABA), and shows a significant association with depression [43,44]. Thus, the LcS intervention for 9 weeks increased the beneficial *Adlercreutzia*, *Megasphaera* and *Veillonella* and decreased the bacteria related to mental illness, such as *Rikenellaceae_RC9_gut_group*, *Sutterella* and *Oscillibacter*.

The IL-1β, IL-6, and TNF-α proinflammatory cytokines were previously reported as elevated in patients with depression and appear to play significant roles in the pathogenesis of major depression [45]. Moreover, these cytokines influenced the intestinal barrier integrity and hypothalamic–pituitary adrenal axis activity and tryptophan metabolism in the kynurenine pathway [45]. IL-6 released at the periphery and/or in the central nervous system exerted effects in the behavioral response to lipopolysaccharide (LPS) and IL-1 that developed in response to systemic inflammation [46]. In this study, the reductions in proinflammatory cytokines may be consistent with the reductions in the proinflammatory bacteria *Sutterella*.

Our study had several limitations. First, no significant differences were observed in the BDI and HAMD scores between groups. The beneficial effects of LcS on depression were not clearly demonstrated, although reductions in the BDI and HAMD scores were observed in the LcS when group compared with the placebo group. Second, our study lacked a metabolite analysis of the intestinal microbiota. Therefore, the precise LcS mechanism remains unknown, and further intestinal microbiota metabolite studies are required to clarify these mechanisms.

## 5. Conclusions

By consuming LcS daily for 9 weeks, constipation and depressive symptoms were significantly improved in patients with depression, although the changes in depressive symptoms were not significant between groups. Additionally, the IL-6 levels were significantly lower in the LcS group than in the placebo group. Additionally, the LcS intervention increased the beneficial *Adlercreutzia*, *Megasphaera* and *Veillonella* levels and decreased the bacteria related to mental illness, including *Rikenellaceae_RC9_gut_group*, *Sutterella* and *Oscillibacter*. Thus, the daily consumption of LcS for 9 weeks relieved constipation and improved potentially depressive symptoms and appeared to regulate the intestinal microbiota associated with mental illness.

## Figures and Tables

**Figure 1 nutrients-13-02238-f001:**
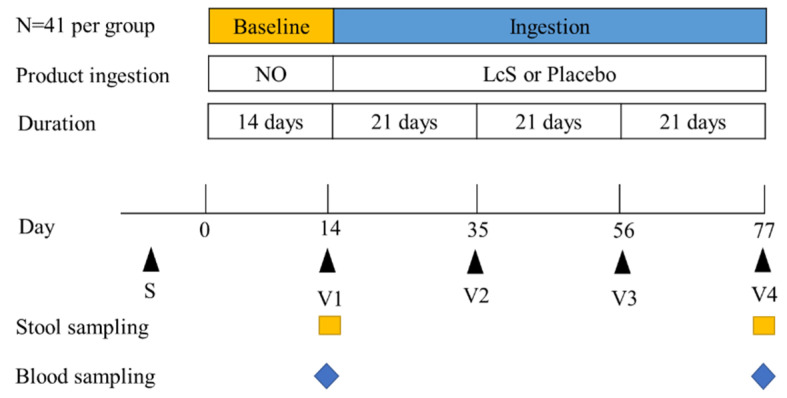
Study design. LcS: the fermented dairy beverage contained *Lacticaseibacillus paracasei* YIT 9029, V1–V4: visit 1–visit 4.

**Figure 2 nutrients-13-02238-f002:**
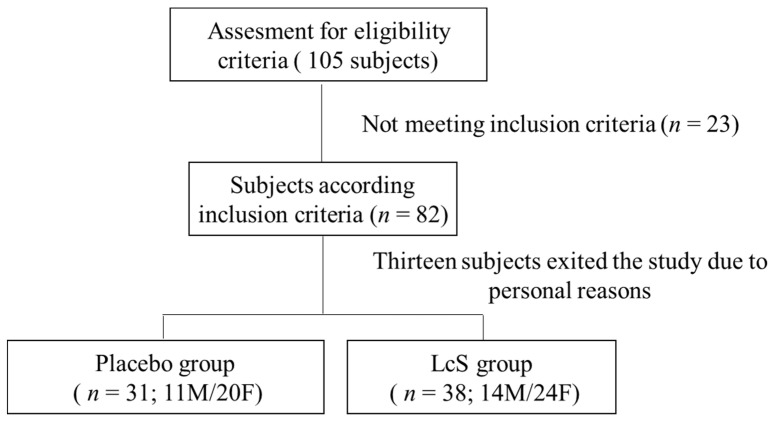
CONSORT study flow chart. M: male; F; female.

**Figure 3 nutrients-13-02238-f003:**
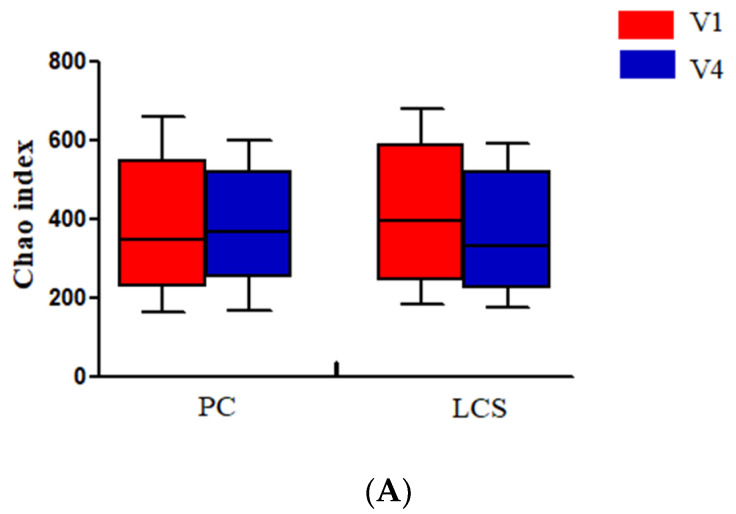
Diversity of the gut microbiota at the operational taxonomic unit (OTU) level. Alpha diversity was evaluated based on the Chao (**A**) and Shannon (**B**). Principal coordinates analysis (PCoA) of the beta diversity was based on the Bray-Curtis. (**C**) Differences between V1 and V4 in the LcS group. (**D**) Differences between V4 in the LcS and placebo groups. (**E**) Differences between V1 and V4 in the placebo group. V1_LCS; samples at V1 in the LcS group, V4_LCS: samples at V4 in the LcS group, V1_PC: samples at V1 in the placebo group, and V4_PC: samples at V4 in the placebo group.

**Figure 4 nutrients-13-02238-f004:**
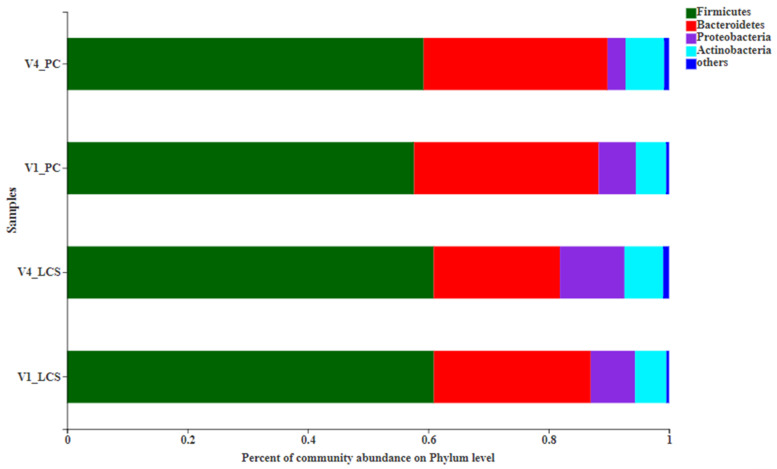
Relative abundance of main phyla in different groups. V1_LCS: samples at V1 in the LcS group, V4_LCS: samples at V4 in the LcS group, V1_PC: samples at V1 in the placebo group, and V4_PC: samples at V4 in the placebo group.

**Figure 5 nutrients-13-02238-f005:**
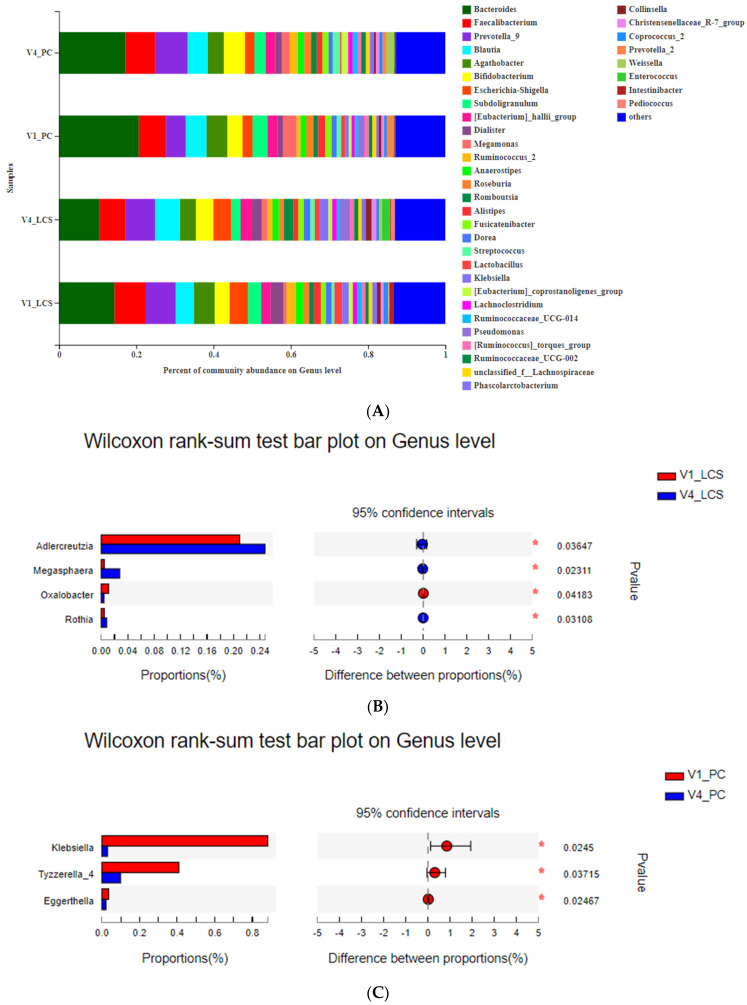
The effects of LcS on the gut microbiota on the genus level. (**A**) The relative abundance of the main genera was >1% in the different groups. (**B**) Significantly different genera levels between V1 and V4 of the LcS group analyzed by the Wilcoxon rank-sum test. (**C**) Significantly different genera levels between V1 and V4 of the placebo group analyzed by the Wilcoxon rank-sum test. (**D**) Significantly different genera levels between V4 of the LcS and placebo groups analyzed by the Wilcoxon rank-sum test. (**E**) Linear discriminant analysis (LDA) integrated with the effect size (LEfSe) analysis indicating genera that were significantly differentially abundant between V4 of the LcS and placebo groups (LDA score ≥ 2). * *p* < 0.05 and ** *p* < 0.01. V1_LCS: samples at V1 in the LcS group, V4_LCS: samples at V4 in the LcS group, V1_PC: samples at V1 in the placebo group, and V4_PC: samples at V4 in the placebo group.

**Table 1 nutrients-13-02238-t001:** Baseline subject demographics.

Groups		Placebo	LcS
Age (years)		49.7 ± 9.6	45.8 ± 12.3
Weight (kg)		66.0 ± 11.8	66.0 ± 10.7
BMI ^a^ (kg/m^2^)		25.0 ± 4.6	24.2 ± 3.4
HAMD ^b^ (score)		17.1 ± 4.4	16.4 ± 4.8
Gender (*n*)	Female	20	24
	Male	11	14

^a^ BMI: Body mass index; ^b^ HAMD: Hamilton depression rating scale. LcS: the fermented dairy beverage contained *Lacticaseibacillus paracasei* YIT 9029.

**Table 2 nutrients-13-02238-t002:** The effects of LcS supplementation on patient assessments of constipation-symptom (PAC-SYM) scores.

Item		Placebo (V1)	LcS (V1)	Placebo (V4)	LcS (V4)	*p*-Value ^a^
Abdominal symptoms	1. Discomfort in abdomen	1.03 ± 0.98	1.05 ± 0.84	0.65 ± 0.71 ^#^	0.45 ± 0.65 *	0.171
	2. Pain in abdomen	0.81 ± 0.83	0.92 ± 0.85	0.48 ± 0.68	0.34 ± 0.53 *	
	3. Bloating in abdomen	0.90 ± 0.87	1.18 ± 0.95	0.71 ± 0.78	0.50 ± 0.65 *	
	4. Stomach cramps	0.74 ± 0.86	0.66 ± 0.97	0.58 ± 0.92	0.34 ± 0.53 *	
Rectal symptoms	5. Painful bowel movement	0.90 ± 1.08	1.18 ± 1.27	0.61 ± 0.72	0.47 ± 0.73 *	0.145
	6. Rectal burning during or after a bowel movement	1.23 ± 1.18	1.21 ± 1.19	0.58 ± 0.67 ^#^	0.42 ± 0.68 *	
	7. Rectal tearing or bleeding after a bowel movement	1.32 ± 1.30	1.11 ± 1.16	0.58 ± 0.76 ^#^	0.29 ± 0.65 *^&^	
Stool symptoms	8. Incomplete bowel movement, like you didn’t “finish”	1.32 ± 1.17	1.26 ± 0.92	1.00 ± 1.06	0.55 ± 0.65 *	0.033
	9. Bowel movement that were too hard	1.77 ± 1.56	1.53 ± 1.31	0.90 ± 0.94 ^#^	0.71 ± 0.80 *	
	10. Bowel movement that were too small	1.55 ± 1.43	1.45 ± 1.41	0.84 ± 0.82 ^#^	0.50 ± 0.65 *	
	11. Straining or squeezing to try to pass bowel movements	1.94 ± 1.39	1.58 ± 1.24	1.16 ± 0.86 ^#^	0.76 ± 0.63 *^&^	
	12. Feeling like you had to pass a bowel movement you couldn’t	1.48 ± 1.34	1.26 ± 1.18	0.87 ± 0.92 ^#^	0.53 ± 0.69 *	
Total scores		15.00 ± 10.39	14.39 ± 9.67	8.97 ± 7.87 ^#^	5.87 ± 6.05 *	0.055

* *p* < 0.05, comparisons with V1 values in the LcS group using the Wilcoxon signed-rank test; ^#^ *p* < 0.05, comparisons with VI values in the placebo group using the Wilcoxon signed-rank test; ^&^ *p* < 0.05, significant differences between the LcS group versus the placebo group at V4 using the Wilcoxon rank-sum test. ^a^
*p* < 0.05, significant differences in the subscales between the LcS group versus the placebo group at V4 using the Wilcoxon rank-sum test. LcS: the fermented dairy beverage contained Lacticaseibacillus paracasei YIT 9029, V1–V4: visit 1–visit 4.

**Table 3 nutrients-13-02238-t003:** The effects of the LcS and placebo on depression.

Item		Placebo (V1)	LcS (V1)	Placebo (V4)	LcS (V4)
BDI	Mean ± SD	14.00 ± 5.53	13.16 ± 7.24	7.39 ± 7.57 ^#^	4.82 ± 6.42 *
	Not depressed individual (%)	3 (9.7)	7 (18.4)	17 (54.8)	24 (63.2)
	Depressed individual (%)	28 (90.3)	29 (81.6)	14 (45.2)	14 (36.8)
	Mild individual (%)	2 (6.5)	2 (5.3)	0 (0)	4 (10.5)
	Moderate individual (%)	15 (48.4)	16 (42.1)	8 (25.8)	7 (18.4)
	Severe individual (%)	11 (35.4)	13 (34.2)	6 (19.4)	3 (7.9)
HAMD	Mean ± SD	17.13 ± 4.43	16.39 ± 4.84	5.84 ± 6.14 ^#^	4.13 ± 2.86 *
	Not depressed individual (%)	0 (0.0)	0 (0.0)	26 (83.9)	34 (89.5)
	Depressed individual (%)	31 (100.0)	38 (100.0)	5 (16.1)	4 (10.5)
	Mild individual (%)	19 (61.3)	25 (65.8)	4 (12.9)	4 (10.5)
	Moderate individual (%)	10 (32.3)	11 (28.9)	0 (0.0)	0 (0.0)
	Severe individual (%)	2 (6.4)	2 (5.3)	1 (3.2)	0 (0.0)

* *p* < 0.05, comparisons with V1 values in the LcS group; mean values were analyzed using the Wilcoxon signed-rank test, and degrees of depression were analyzed using McNemar’s test; ^#^ *p* < 0.05, comparisons with V1 values in the placebo group; mean values were analyzed using the Wilcoxon signed-rank test, and degrees of depression were analyzed using McNemar’s test. Value representative for the proportion of depression subjects. BDI: Beck Depression Index, HAMD: Hamilton depression rating scale.

**Table 4 nutrients-13-02238-t004:** The effects of LcS on the proinflammatory cytokines.

Item	Placebo (V1)	LcS (V1)	Placebo (V4)	LcS (V4)
IL-1β (pg/mL)	24.73 ± 0.59	24.88 ± 1.26	15.33 ± 5.31 ^#^	15.67 ± 6.97 *
IL-6 (pg/mL)	7.56 ± 2.28	7.68 ± 1.97	8.39 ± 3.58	6.82 ± 1.21 *^&^
TNF-α (pg/mL)	19.26 ± 2.09	21.16 ± 3.70	16.22 ± 1.67 ^#^	17.19 ± 3.06 *

* *p* < 0.05, comparison with V1 values in the LcS group; ^#^ *p* < 0.05, comparison with V1 values in the placebo group; ^&^
*p* < 0.05, significant differences between the LcS and placebo groups. IL: interleukin, TNF: tumor necrosis factor.

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
