# Peer review of "Effects of Fermented Milk Containing Lacticaseibacillus paracasei Strain Shirota on Constipation in Patients with Depression: A Randomized, Double-Blind, Placebo-Controlled Trial"

_nutrients, 2021, doi:10.3390/nu13072238_

Round 1

Reviewer 1 Report

General comments

=============

This paper is a randomized placebo-controlled trial, assessing the effects of LcS on constipation and depressive symptoms in patients with depression. I recommend the following points are addressed before it is considered for publication:

Specific comments

=============

Major comments

  1. The authors should clarify whether they administered antidepressants to both groups (LcS group and placebo group) during the study period. If they did not administer them, I would question the following points: 1) did none of the patients in both groups worsen in the depressive state or drop out during the study period while patients in both groups were moderately depressed at baseline? 2) why did the depressive state (the BDI and HAMD scores) improve significantly in the placebo group at the endpoint?

Minor comments

  1. In the material and methods section (2.2.), the authors should explain the details of the Rome IV criteria, and the rationale for using the cut-off score (8 scores) in HAMD.
  2. In the material and methods section (2.4.), could the authors show the data of alpha diversity even if it was not significant?
  3. In the material and methods section (2.5.), could the authors explain why they selected the three biomarkers?
  4. In Table 2 and Table 3, it would be better to move the footnotes of the PAC-SYM, BDI scores, and HAMD-17 scores to the material and methods section.
  5. In the discussion section (Line. 322), could the authors provide evidence that Sutterella is a pro-inflammatory bacteria?

Author Response

Response to Reviewer 1 Comments

Major comments

  1. The authors should clarify whether they administered antidepressants to both groups (LcS group and placebo group) during the study period. If they did not administer them, I would question the following points: 1) did none of the patients in both groups worsen in the depressive state or drop out during the study period while patients in both groups were moderately depressed at baseline? 2) why did the depressive state (the BDI and HAMD scores) improve significantly in the placebo group at the endpoint?

Response 1: During the study period, Subjects did not administered antidepressants, and none of the patients in both groups worsen in the depressive state. The reason for depressive state (the BDI and HAMD scores) improving significantly in the placebo group at the endpoint may be due to two factors: one was the placebo response, another was the placebo containing milk content, which was reported that positive health and mood changes were associated with the consumption of raw milk in a consumer survey. The reasons were also revised in the discussion section and the contexts are as follows:

Another reason for depressive state (the BDI and HAMD scores) improving significantly in the placebo group was the placebo containing milk content, which was reported that positive health and mood changes were associated with the consumption of raw milk in a consumer survey.

Minor comments

  1. In the material and methods section (2.2.), the authors should explain the details of the Rome IV criteria, and the rationale for using the cut-off score (8 scores) in HAMD.

Response: The details of the Rome IV criteria, and the rationale for using the cut-off

score (8 scores) in HAMD were added in the section (2.2), and the contexts are as

follows:

a constipation diagnosis according to the Rome IV Criteria, including two or more symptoms of the following: (i) straining more than 25% of defecations; (ii) lumpy or hard stools more than 25% of defecations; (iii) sensation of incomplete evacuation more than one-fourth (25%) of defecations; (iv) sensation of anorectal obstruction/blockage more than one-fourth (25%) of defecations; (v) manual maneuvers to facilitate more than one-fourth (25%) of defecations; (vi) fewer than three spontaneous bowel movements per week. In addition, Hamilton depression rating scale (HAMD) 17 items score of ≥ 8 is considered depression.

  1. In the material and methods section (2.4.), could the authors show the data of alpha diversity even if it was not significant?

Response 2: The data of alpha diversity was supplemented in the material and methods section (2.4.) and results section (3.4.). The contexts are as follows:

In the material and methods section (2.4.): Chao and Shannon indices were calculated to assess alpha diversity.

In the results section (3.4.): The α-diversity of gut microbiota at the OTU level was identified, the OTU numbers and the Chao index or Shannon index was not significantly different among the two groups (Figure 3A and B).

  1. In the material and methods section (2.5.), could the authors explain why they selected the three biomarkers?

Response 3: The reason for choosing the three biomarkers was shown in the discussion section and the contexts are as follows:

The IL-1β, IL-6, and TNF-α pro-inflammatory cytokines were previously reported as elevated in patients with depression, and appeared to play significant roles in the pathogenesis of major depression [45].

  1. In Table 2 and Table 3, it would be better to move the footnotes of the PAC-SYM, BDI scores, and HAMD-17 scores to the material and methods section.

Response 4: The footnotes of the PAC-SYM, BDI scores, and HAMD-17 scores were moved to the material and methods section (2.3.)

  1. In the discussion section (Line. 322), could the authors provide evidence that Sutterella is a pro-inflammatory bacteria?

Response 5: In the discussion section, the evidence of pro-inflammatory Sutterella was supplemented, and the contexts are as follows:

Sutterella was widely prevalent commensals with mild pro-inflammatory capacity in the human gastrointestinal tract.

Reviewer 2 Report

The manuscript submitted by Zhang et al., was well written and easy to understand. They performed a randomized double-blind placebo controlled clinical trial on patients with comorbid depression and constipation using the probiotic strain Lactobacillus casei Shirota (YIT 9029). The intervention was taken every day for 9 weeks and patients filled out daily diaries on intervention adherence, dietary details, defecation details, mood details, and medications. Additionally, pre- and post-treatment fecal microbiome and blood serum measures were observed.  

 Major considerations:

  1. It is unclear what the placebo was. It is stated on line 133 that the placebo was “Yakult (China) Corporation.” Was this provided by Yakult China and it was the Yakult drink without LcS (YIT 9029)? This is an important detail because the placebo was essentially as effective as the treatment and both were effective in reducing both depression and GI distress. Therefore, it is natural for a reader to try to determine what the placebo was in order to figure out why it was equally as effective as the treatment.
  2. There were very few results where the intervention and the placebo differed in their effectiveness. It seems like many of the results were overstated in that it was highlighted that LcS alleviated a symptom, but it is equally correct to state that the placebo alleviated the same symptom.
  3. Wilcoxon Rank Sum is an inappropriate analysis for comparing individual taxa as was done in Figure 5. Analysis of Compositions of Microbiomes (ANCOM) is an appropriate analysis for comparing taxa. ANCOM is more adept in preventing type 1 error. Much of the discussion focuses on individual taxa differences and will likely change dramatically after ANCOM is performed.
  4. The cytokine levels in Table 4 seem very high. Normal reference levels for IL-6 and TNFa are less than 1.8 pg/ml and 2.8 pg/ml respectively. Additionally, another publication by Vara et al., 2018 (doi: 2147/IJGM.S166600) showed patients with IBS with substantially lower concentrations of IL-6, IL-1B, and TNFa.
  5. Claims around the microbiome and depression were not sufficiently supported nor substantiated. It was stated on line 60-64 that depressed patients showed lower bifidobacterial and lactobacilli. However, there have been several reviews on this topic and it is misleading not to present more evidence to support only citing one study on this topic. Additionally, the final statement of the paper on line 340 is simply not supported by the results of this study.

Minor considerations:

  1. Line 84-85 it is unclear how levels of butyrate in stool is linked to improving constipation.
  2. Bray-Curtis is not an analysis technique. It is a distance metric. The beta diversity analysis tool needs to be stated.
  3. Line 237 states that “We observed differences in overall microbial composition between both groups (Figure 3).” However, based on the stats presented in Figure 3 there were no significant differences.
  4. Figure 4 panel A should be revised to include only the top 15 or 20 genera and the remaining low abundance genera should be consolidated into the “others” category.

Author Response

Response to Reviewer 2 Comments

Major considerations:

  1. It is unclear what the placebo was. It is stated on line 133 that the placebo was “Yakult (China) Corporation.” Was this provided by Yakult China and it was the Yakult drink without LcS (YIT 9029)? This is an important detail because the placebo was essentially as effective as the treatment and both were effective in reducing both depression and GI distress. Therefore, it is natural for a reader to try to determine what the placebo was in order to figure out why it was equally as effective as the treatment.

Response 1: The information of test drinks was supplemented in the material and methods section (2.3.), and the contexts are as follows:

The test drinks were a fermented dairy beverage and a placebo. Each 100-mL bottle of the fermented dairy beverage contained at least 1.0 × 1010 CFU of Lacticaseibacillus paracasei YIT 9029 (strain Shirota: LcS), water, sugar, skimmed milk powder, glucose, and flavor. The nutrient composition of one bottle was 287 kJ of energy, 1.2 g of protein, 0 g of lipid, 15.7 g of carbohydrate, and 19 mg of sodium. The placebo contained the same components as the fermented dairy beverage but did not contain any bacteria. The taste of the placebo was made similar to that of the fermented dairy beverage by adding lactic acid. The taste, flavor, and external appearance of the placebo were indistinguishable from those of the fermented dairy beverage. The fermented dairy beverage and the placebo were both manufactured and distributed by Yakult (China) Corporation, Shanghai, China. They were refrigerated until ingestion.

The reason for depressive state (the BDI and HAMD scores) improving significantly in the placebo group at the endpoint may be due to two factors: one was the placebo response, another was the placebo containing milk content, which was reported that positive health and mood changes were associated with the consumption of raw milk in a consumer survey. The reasons were also revised in the discussion section and the contexts are as follows:

Another reason for depressive state (the BDI and HAMD scores) improving significantly in the placebo group was the placebo containing milk content, which was reported that positive health and mood changes were associated with the consumption of raw milk in a consumer survey.

  1. There were very few results where the intervention and the placebo differed in their effectiveness. It seems like many of the results were overstated in that it was highlighted that LcS alleviated a symptom, but it is equally correct to state that the placebo alleviated the same symptom.

Response 2: The effectiveness of LcS intervention was shown in improving PAC-SYM scores especially stool symptoms and decreasing the level of IL-6. In the Abstract and Conclusion section, we tried to describe the result by distinguishing the ones showed a difference between groups from those without difference between groups. The reason for depressive state (the BDI and HAMD scores) also improving significantly in the placebo group at the endpoint may be due to two factors: one was the placebo response, another was the placebo containing milk content, which was reported that positive health and mood changes were associated with the consumption of raw milk in a consumer survey.

  1. Wilcoxon Rank Sum is an inappropriate analysis for comparing individual taxa as was done in Figure 5. Analysis of Compositions of Microbiomes (ANCOM) is an appropriate analysis for comparing taxa. ANCOM is more adept in preventing type 1 error. Much of the discussion focuses on individual taxa differences and will likely change dramatically after ANCOM is performed.

Response 3: Wilcoxon Rank Sum was used to comparing individual taxa in many publications, for example , Peng et al., (2020 doi: 10.3389/fmicb.2019.03141) used Wilcoxon rank-sum tests to compare differences in fecal bacterial communities between Goto-Kakizaki (GK) rat and Wistar rats at the genus level.

ANCOM and LEfSe were considered non-parametric (assumes no underlying distribution of data), and consistently resulted in higher concordances (Zachary et al., 2021, doi.org/10.1186/s12859-021-04193-6), so we performed LEfSe analysis to compare the differences in genera between V4 of the LcS and placebo groups. The results were similar with that analyzed by Wilcoxon Rank Sum, and supplemented in the results section (3.4), and the contexts are as follows:

LEfSe was used to further determine whether specific bacterial taxa were differentially enriched in V4 of LcS group and placebo group. Using a logarithmic LDA score cutoff of 2, 17 discriminatory genera were identified as key discriminants (Figure 5E). Veillonella, Neisseria, Ralstonia and Eggerthella were significantly overrepresented in the feces of LcS group, whereas Rikenellaceae_RC9_gut_group, Sutterella, Oscillibacter and Rothia were enriched in placebo group.

  1. The cytokine levels in Table 4 seem very high. Normal reference levels for IL-6 and TNFa are less than 1.8 pg/ml and 2.8 pg/ml respectively. Additionally, another publication by Vara et al., 2018 (doi: 2147/IJGM.S166600) showed patients with IBS with substantially lower concentrations of IL-6, IL-1B, and TNFa.

Response 4: The cytokine levels in Table 4 were not high. In many publications, the

cytokine levels were similar with that in our study, for example Nikolaos et al., 2005 (DOI: 10.3748/wjg.v11.i11.1639) showed normal individuals with 0.45-9.96 pg/mL of IL-6, colorectal cancer patients with 1.09-188.42 pg/mL of IL-6, and normal individuals with 0-4.71 pg/mL of TNFa, colorectal cancer patients with 1.46-292.13 pg/mL of TNFa.

  1. Claims around the microbiome and depression were not sufficiently supported nor substantiated. It was stated on line 60-64 that depressed patients showed lower bifidobacterial and lactobacilli. However, there have been several reviews on this topic and it is misleading not to present more evidence to support only citing one study on this topic. Additionally, the final statement of the paper on line 340 is simply not supported by the results of this study.

Response 5: Microbiota are important in normal healthy brain function. Several studies

have shown that microbiota influence behavior and that immune challenges that influence anxiety- and depressive-like behaviors are associated with alterations in microbiota (Foster et al., 2013, doi: 10.1016/j.tins.2013.01.005. ). We added a more evidence to support the topic and the contexts are as follows:

Kunugi et al. reported results suggesting that individuals with lower bacterial counts for Bifidobacterium and/or Lactobacillus are more common in patients with major depressive disorder than in healthy controls.

Minor considerations:

  1. Line 84-85 it is unclear how levels of butyrate in stool is linked to improving constipation.

Response 1: The contexts are revised to “LcS also improved hard stools during constipation, possibly by increasing butyrate-producing bacteria and subsequent stool butyrate levels by 39.7%.”

  1. Bray-Curtis is not an analysis technique. It is a distance metric. The beta diversity analysis tool needs to be stated.

Response 2: In the results section 3.4, the contexts are revised to “The β-diversity of gut microbiota at the OTU level was identified using PCoA based on Bray-Curtis distances”.

  1. Line 237 states that “We observed differences in overall microbial composition between both groups (Figure 3).” However, based on the stats presented in Figure 3 there were no significant differences.

Response 3: In the results section 3.4, the contexts are revised to “The overall microbial composition was observed between both groups (Figure 3)” 

  1. Figure 4 panel A should be revised to include only the top 15 or 20 genera and the remaining low abundance genera should be consolidated into the “others” category.

 Response 4: Do you mean Figure 5 panel A? We have supplemented the sentence in the results section 3.4, and the contexts are as follows:

Bacteria with a relative abundance of more than 1% were analyzed at the genus level (Figure 5A).

Round 2

Reviewer 1 Report

None.